

# Indirect Doppler flow systolic blood pressure measurements taken with and without headphones in privately-owned, conscious dogs

India R. Gill[1], Joshua M. Price[2] and Jacqueline C. Whittemore[1]

[1] Department of Small Animal Clinical Sciences, University of Tennessee—Knoxville, Knoxville, TN, United States of America

[2] Office of Information Technology, University of Tennessee—Knoxville, Knoxville, TN, United States of America

## ABSTRACT

**Objective**. The purpose of this study was to assess the effect of headphone use and covariates on indirect radial Doppler flow systolic arterial blood pressure (BP) measurements in dogs.

**Methods**. Between May and August 2018, 100 privately-owned dogs were enrolled. Blood pressure was measured in lateral recumbency, with and without headphones, using a randomized crossover design. The initial BP, mean of BP 2-6, weight, BCS, MCS, anxiety score, and heart rate were recorded. Mixed effects crossover analyses and Spearman rank correlation coefficients were determined.

**Results**. Eighty-four dogs completed the study. Eleven dogs were removed due to excessive anxiety, 10 of which were in the non-headphone first group. The number of dogs diagnosed as hypertensive did not differ between measurement types (19 vs. 18), with seven dogs categorized as hypertensive during both periods. Significant differences in BP were identified ($F[1, 80] = 4.3$, $P = 0.04$) due to higher results for measurements taken without headphones for BP 1, but not BP 2-6. Systolic BP was positively correlated with anxiety score, age, and weight.

**Conclusions and Clinical Relevance**. Though BP 1 was significantly higher when taken without headphones, this pattern did not persist for BP 2-6. Lack of association between BP 2-6 results and measurement type could reflect exclusion of dogs most sensitive to white coat hypertension, acclimation to technique, or improved sound quality of headphones. Given significantly higher BP 1 results and disproportionate exclusion of dogs due to anxiety when measurements first were taken without headphones, use of headphones is recommended to improve accuracy of results.

## INTRODUCTION

Excitement or anxiety associated with the process of veterinary evaluation and blood pressure measurement can activate the central nervous system and increase systolic blood pressure, a phenomenon known as the white-coat effect or white-coat hypertension (*Belew, Barlett & Brown, 1999*; *Bragg et al., 2015*; *Höglund et al., 2012*). Habituation has been

Corresponding author
Jacqueline C. Whittemore,
jwhittemore@utk.edu

associated with mixed effects on blood pressure measurements in cats. Although there was no significant difference in white-coat effect among five simulated office visits for healthy research cats undergoing repeated indirect oscillometric blood pressure measurement after physical examination, the magnitude of the white-coat effect was lowest during the first visit for 3/6 cats (*Belew, Barlett & Brown, 1999*). Failure to recognize the white-coat effect can result in unnecessary life-long treatment (*Acierno et al., 2018*) and potentially result in iatrogenic systemic hypotension. Conversely, failure to detect true hypertension can result in delayed diagnosis and, thus, progression of end-organ damage. As such, accurate blood pressure determination is essential to identify the presence of hypertension and need for intervention (*Höglund et al., 2012*).

The American College of Veterinary Internal Medicine has developed and recently updated guidelines for appropriate collection of indirect blood pressure measurements (*Acierno et al., 2018*). These include acclimating the patient to the room, equipment, and personnel; taking care in proper patient positioning and crystal placement; using an appropriately sized cuff and discarding outlier values; and having an experienced individual collect repeated measurements (*Rondeau, Mackalonis & Hess, 2013*). Some professionals recommended that headphones be used for collection of measurements using the indirect Doppler flow technique lest noise from the machine startle the patient (*Caney, 2007*; *Whittemore, Nystrom & Mawby, 2017*). However, measurements are routinely taken without headphones as the importance of this recommendation has not been confirmed. One abstract reported no significant differences in indirect Doppler flow measurement results collected with and without headphones 6 weeks apart in cats (*Williams, Elliot & Syme, 2010*). Data regarding the impact of headphone use on measurements taken in dogs are lacking.

The purpose of this study was to assess the impact of the use of headphones on indirect systolic blood pressure measurements taken using the Doppler method in privately-owned dogs. We hypothesized that indirect blood pressure measurements taken with and without the use of headphones would be discordant, with higher results for measurements taken without headphones.

## MATERIALS & METHODS

### Study population

This study was conducted at the University of Tennessee's Veterinary Medical Center and was approved by the Institutional Animal Care and Use Committee of the University of Tennessee, Knoxville (protocol number 2428).

Privately-owned dogs were enrolled in the study between May and August of 2018. Dogs belonging to faculty, staff, and students of the College, as well as patients of the Veterinary Medicine Center, were eligible for enrollment. Prior to enrollment, informed consent was obtained for each dog from the owner, along with information regarding known medical conditions and current medications.

Dogs that were intolerant of the procedure, aggressive, or received sedation or anesthesia within the previous 12 h were excluded from the study. All other dogs, regardless of disease

status or medications received, were included to best approximate a clinical population and allow comparison with prior studies (*Bosiack et al., 2010*; *Hsiang, Lien & Huang, 2008*; *Mooney et al., 2017*; *Wernick et al., 2012*).

### Randomization

A randomization table was generated using a randomized number generator (https://www.random.org, accessed 5/14/18) to determine whether subjects would undergo blood pressure measurement with or without headphones first.

### Data collection

All data collection was performed in a quiet examination room by 1 investigator (IRG). If the owner was unable to be present for the duration of the protocol, the process was either carried out by the investigator alone or with the help of a veterinary assistant who was present for the duration of the visit.

Weight, body condition score (BCS) and muscle condition score (MCS) (*Baldwin et al., 2010*), anxiety score (*Scansen et al., 2014*), and heart rate were recorded immediately upon entry into the room. The original American College of Veterinary Internal Medicine Consensus Statement guidelines (*Brown et al., 2007*) were followed for collection of blood pressure measurement, because the current guidelines (*Acierno et al., 2018*) had not been published at the time of study completion. Measurements were taken using the left radial artery with the dog in right lateral recumbency. If this position could not be used due to absence of a limb, injury, pain, or intravenous catheter location, the right forelimb was used with the animal in left lateral recumbency.

A soft measuring tape was used to measure the circumference of the mid-antebrachium, and a cuff was selected such that the width of the cuff was 30–40% of the limb circumference at the site of cuff placement. The palmar surface of the foot between the carpal and metacarpal pads was shaved for placement of the Doppler crystal. The dog then was allowed to acclimate to the environment and personnel for 10 min. Heart rate and anxiety score were recorded at the beginning and end of the acclimation period, as were acclimation start and stop times.

After the conclusion of the acclimation period, the investigator placed a blood pressure cuff on the mid-antebrachium with the dog in lateral recumbency and the forelimb held at the level of the heart. Headphones were worn by the investigator when indicated by the randomization system. The concave side of a flat infant Doppler probe was covered in ultrasonic coupling gel and secured over the artery with bandage tape. The cuff was inflated to approximately 20 mmHg above the point at which blood flow could no longer be heard. Air was slowly released until blood flow was audible, the value of which was recorded. Five more readings were obtained in this manner. Heart rate and anxiety score were collected at the start and stop of each measurement series, as were measurement start and stop times. Following the first set of data collection, all equipment was removed and a second 10-minute acclimation period started, with the procedure repeated with or without headphones as per the crossover design.

The same Doppler unit and probe (Model 811-B Doppler ultrasonic Flow Detector with flat infant probe; Parks Medical Electronics, Inc., Aloha, OR, USA), sphygmomanometer

(Riester Ri-san® aneroid sphygmomanometer; Riestar Direct, Ventura, CA, USA), blood pressure cuffs (SunTechMed Soft Disposable Cuffs, Suntech Medical, Morrisville, NC, USA), and headphones (Panasonic RP-HT161 Stereo Headphones, Suntech Medical, Morrisville, NC, USA) were used for all subjects.

## Data entry and statistical analyses

After each visit, data collection forms were submitted to a person unrelated to the study. This individual entered the data into a file that was unavailable to investigators until the conclusion of data collection.

Descriptive statistics were performed for each response measure. Continuous measures were analyzed for normality using the Shapiro–Wilk test and Q-Q plots. Each continuous measure was screened for outliers using box-and-whisker plots. Equality of variances was evaluated using Levene's test for equality of variances. Parameters with normally distributed data were reported as mean ± standard deviation, with non-normal data reported as median (range).

The number of dogs categorized as hypertensive based on blood pressure measurements >150 mmHg (*Brown et al., 2007*) was determined for each measurement type and period. The paired sample student's $t$-test was used to compare the time required to complete data collection for measurements collected with vs. without headphones.

Spearman rank correlation coefficients were used to assess correlations among BP 1, BP 2–6, and possible covariates (age, weight, BCS, MCS, cuff size as a percent of limb circumference; heart rate and anxiety score at the start of the measurement series; and time required for collection of measurements) for each measurement type (headphone vs. non-headphone). Spearman rank correlation coefficients and variance inflation factors from regression analysis were used to assess for collinearity between possible covariates prior to performing ANCOVA models.

A mixed effects crossover design with corresponding ANCOVAs was performed to determine if BP 1 and BP 2-6 differed between headphone and non-headphone measurements and to assess whether the washout period between measurement techniques was adequate (*Littell et al., 2006*). Period and measurement type (headphone vs. non-headphone) were included as fixed effects. Continuous measures with $\rho > 0.2$ on correlation testing were included as covariates in the initial analysis. Covariates initially included in the ANCOVA models were age, BCS, MCS, weight, as well as anxiety score and heart rate taken at the start of measurements. Dog nested within sequence was included as a random effect. A compound symmetry variance/covariance structure was incorporated into each model to account for constant covariates. Each covariate was tested for homogeneity of slopes between measurement types. Equal slopes ANCOVA models were determined for each covariate (*Littell et al., 2006*). Backwards variable selection was performed on the full models to determine which covariates explained significant variability in results. Covariates included in the final models were period, measurement type (headphone vs. non-headphone), age, weight, and anxiety score. The Shapiro–Wilk test of normality of the residuals and Levene's test for equality of variances was evaluated to ensure the assumptions of the statistical method had been met. Commercial statistical software packages (SAS 9.4
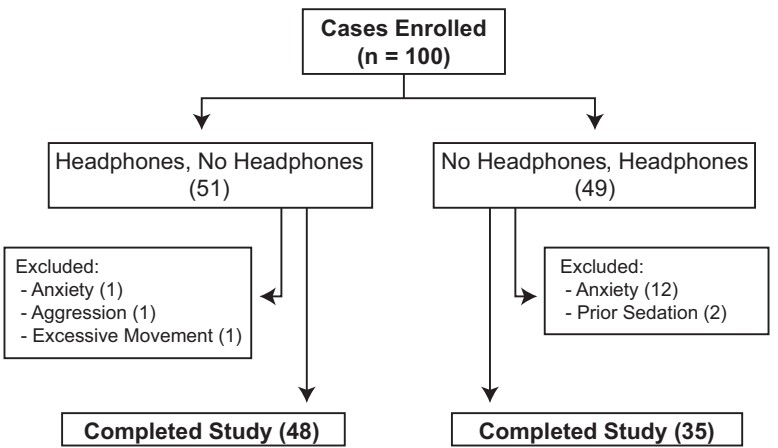

**Figure 1** Flowchart showing the distribution of 100 privately-owned dogs randomized to undergo collection of systolic blood pressure measurements with and without headphones using a crossover design.

release TS1M5; SAS Institute Inc., Cary, NC, USA; IBM SPSS Statistics for Windows, version 25; IBM Corp., Armonk, NY, USA) were used for all analyses. $P < 0.05$ was considered significant.

## RESULTS

One hundred dogs initially were enrolled in the study (File S1), 17 of which were excluded (Fig. 1). Of the remaining 83 dogs (Table 1), 48 dogs had measurements taken using headphones first and 35 dogs had measurements taken without headphones first. Demographic information and results of each blood pressure measurement series are listed in Table 1, while information for each dog is presented in File S1. The most common breeds included were mixed breed dogs (39), Golden retriever (6), Labrador retriever (6), and Border Collie (3) with ≤ 2 dogs of 27 other breeds. Fifty-eight dogs were categorized as healthy. The most common diseases identified in the remaining 25 dogs were categorized as metabolic (13), musculoskeletal (6), dermatologic (5), endocrine (4), and neurologic (3), with 13 dogs having > 1 condition diagnosed. Seven dogs had been diagnosed with ≥ 1 disease associated with hypertension: chronic kidney disease (4), hyperadrenocorticism (2), and diabetes mellitus with uncharacterized azotemia (1). Five dogs were receiving a medication that could potentially affect blood pressure results. Three dogs were receiving prednisone, 1 dog with chronic kidney disease was receiving amlodipine, and a different dog with chronic kidney disease was receiving phenylpropanolamine.

There was no difference in the number of dogs that would have been diagnosed as hypertensive between measurement types (headphones, 19 dogs; without headphones, 18 dogs) or periods (period 1, 21 dogs; period 2, 16 dogs). Seven dogs were categorized as hypertensive during both periods. Time required to collect blood pressure measurements also did not differ between measurement types (headphone, $4.9 \pm 2.3$ min; without headphones, $4.7 \pm 2.1$ min).

**Table 1 Demographics and indirect Doppler systolic arterial blood pressure readings collected with and without headphones using a crossover design for 83 privately-owned dogs.** Values are reported as median ± standard deviation.

|  | Headphones, no headphones | No headphones, headphones |
|---|---|---|
| Sex |  |  |
|    F, FS | 4, 20 | 0, 15 |
|    M, MC | 4, 20 | 3, 17 |
| Age (years) | 5.6 ± 3.4 | 5.16 ± 3.0 |
| BCS | 5.5 ± 1.1 | 5.4 ± 1.2 |
| MCS | 2.9 ± 0.4 | 2.9 ± 0.3 |
| Weight (kg) | 22.5 ± 14.9 | 27.4 ± 22.7 |
| First measurement period |  |  |
|   • Anxiety score | 1.6 ± 0.7 | 1.7 ± 0.9 |
|   • BP 1 (mmHg) | 135 ± 25 | 144 ± 35 |
|   • Mean of BP 2-6 (mmHg) | 134 ± 21 | 141 ± 33 |
| Second measurement period |  |  |
|   • Anxiety score | 1.4 ± 0.7 | 1.6 ± 0.9 |
|   • BP 1 (mmHg) | 136 ± 23 | 135 ± 26 |
|   • Mean of BP 2-6 (mmHg) | 136 ± 24 | 135 ± 27 |

**Notes.**

F, Female; FS, female spayed; M, male; MC, male castrated; BCS, body condition score on a scale of 1–9; MCS, muscle condition score on a scale of 1–3; BP, blood pressure.

Correlation between BP 1 and BP 2-6 was almost perfect for measurements taken with ($\rho = 0.95$, $P < 0.01$) and without ($\rho = 0.94$, $P < 0.01$) headphones. For headphones measurements, the median BP 1 for all dogs across both periods was 130 mmHg (range, 80-210 mmHg) while the median BP 2-6 was 133 mmHg (range, 84–202 mmHg). For non-headphones measurements, the median BP 1 for all dogs across both periods was 136 mmHg (range, 90–270 mmHg) while the median BP 2-6 was 131 mmHg (range, 86–255 mmHg).

With regard to potential covariates, BP 1 was significantly correlated with BCS for measurements taken with headphones ($\rho = 0.24$, $P = 0.03$) but not for measurements taken without headphones. Conversely, BP 2-6 measurements were significantly correlated with age ($\rho = 0.26$, $P = 0.02$) and BCS ($\rho = 0.26$, $P = 0.02$), respectively, for headphones measurements. Similar correlations were not found for BP 2-6 measurements taken without headphones. See Table 2 for significant associations identified among covariables.

Potential covariates removed from the mixed effects model for BP 1 due to failure to explain significant variability in the results were BCS, MCS, and heart rate. Thus, the final model included period, measurement type, age, weight, and anxiety score. There was no significant effect of period. Significant differences in BP 1 results were found between measurement type ($F[1, 80] = 4.3$, $P = 0.04$). Post hoc analysis revealed that differences were due to higher results when taken without headphones (5.4 mmHg increase in the marginal means). Significant covariates affecting BP 1 were anxiety score ($\beta = 9.5$, $t(158) = 3.4$, $P < 0.01$), age ($\beta = 2.4$, $t(83) = 3.1$, $P < 0.01$), and weight ($\beta = 0.4$, $t(83) = 3.4$, $P < 0.01$). Thus, for every 1 unit increase in anxiety score, a 9.5 mmHg increase

**Table 2** Spearman rank correlation coefficients ($\rho$) among covariables for 83 privately-owned dogs in which indirect Doppler systolic arterial blood pressure readings were collected with and without headphones.

|  | Headphones | Without headphones |
| --- | --- | --- |
| **Anxiety score** |  |  |
| Age | $\rho = -0.23, P = 0.04$ | $\rho = -0.26, P = 0.02$ |
| Weight | $\rho = -0.23, P = 0.03$ | $\rho = -0.25, P = 0.02$ |
| Heart rate | $\rho = 0.34, P < 0.01$ | $\rho = 0.38, P < 0.01$ |
| MCS | $\rho = 0.30, P = 0.01$ | $\rho = 0.22, P = 0.04$ |
| Total measurement time | $\rho = 0.25, P = 0.02$ | $\rho = 0.23, P = 0.04$ |
| **Age** |  |  |
| MCS | $\rho = -0.39, P < 0.01$ | $\rho = -0.39, P < 0.01$ |
| Anxiety score | $\rho = -0.23, P = 0.04$ | $\rho = -0.25, P = 0.02$ |
| **Heart rate** |  |  |
| Weight | $\rho = -0.29, P = 0.01$ | $\rho = -0.39, P < 0.01$ |
| MCS | $\rho = 0.30, P = 0.01$ | $\rho = 0.19, P = 0.08$ |
| Anxiety score | $\rho = 0.36, P < 0.01$ | $\rho = 0.38, P < 0.01$ |
| **Total measurement time** |  |  |
| MCS | ns | $\rho = 0.23, P = 0.04$ |
| Anxiety score | $\rho = 0.25, P = 0.02$ | $\rho = 0.23, P = 0.04$ |

**Notes.**
MCS, muscle condition score on a scale of 1–3; ns, not significant.

was observed in BP 1 holding all other measures constant. Similarly, for every year increase in age, BP 1 increased by 2.4 mmHg. For every kg increase in weight, BP 1 increased by 0.4 mmHg.

For BP 2-6, potential covariates removed from the final mixed effects model were BCS, MCS, and heart rate. There was no significant effect of period on the model. Furthermore, there was no significant association between BP 2-6 and headphone use. Significant covariates affecting BP 2-6 were anxiety score ($\beta = 9.7$, $t(158) = 3.6$, $P < 0.01$), age ($\beta = 2.4$, $t(83) = 3.2$, $P < 0.01$), and weight ($\beta = 0.4$, $t(83) = 3.1$, $P < 0.01$). Thus, for every 1 unit increase in anxiety score, a 9.7 mmHg increase was observed in BP 2-6, holding all other measures constant. For every year increase in age, BP 2-6 increased by 2.4 mmHg. Finally, for every kg increase in weight, BP 2-6 increased by 0.4 mmHg.

# DISCUSSION

In this study, no significant difference was identified between BP 2-6 results taken with or without headphones in dogs. However, BP 1 results taken without the use of headphones were significantly higher than results taken with headphones. Although some dogs with high anxiety scores in this study were repeatably normotensive, a 1-point increase in anxiety score was associated with an increase of >9 mmHg in blood pressure overall. Finally, in spite of even randomization between the two groups, 10 times as many dogs in the group randomized to first have measurements taken without the use of headphones were excluded from the study due to excessive anxiety. Thus, it is possible that the lack of association

between headphone usage and BP 2-6 results reflects self-elimination from blood pressure measurement by dogs most vulnerable to white coat effect.

Another factor that might have influenced results is differing speaker quality between the Doppler unit and headphones. It is possible that improved sound clarity, decreased static, and reduction of ambient noise associated with using headphones allowed for earlier detection of the return of blood flow during cuff deflation. This could have offset any mitigating effects of headphone use on changes in pressure due to sounds emitted by the Doppler unit, particularly during patient movement.

The impact of age on blood pressure in dogs remains unclear (*Acierno et al., 2018*). In this study, blood pressure increased 2.4 mmHg for every year of increasing age. These results are consistent with results of some, but not all, prior reports. Differences in associations among studies could reflect differences in study methodology or the population evaluated. For example, two studies identifying a significant positive association between age and blood pressure included age as a continuous variable in the analysis (*Bodey & Michell, 1996*; *Bright & Dentino, 2002*). In contrast, one study that found no association between age and blood pressure did not evaluate age as a continuous variable (*Meurs et al., 2000*). Dogs instead were categorized as adult vs. geriatric based on their age relative to their weight. Further, not all studies included assessment of anxiety or temperament in their analyses. In this study, age was inversely correlated with anxiety score, potentially reflecting an age-related decrease in excitability. Because they were inversely associated, failure to include both in the statistical model could mask associations between blood pressure and either variable. Finally, screening of apparently healthy dogs was not performed in this or prior reports for diseases associated with hypertension, such as chronic hyperadrenocorticism, diabetes mellitus, or kidney disease. In one study that found no association between age and blood pressure, approximately half of enrolled dogs had at least one known disease (*Mooney et al., 2017*), but only 20% of them had a disease known to be associated with hypertension. Positive associations in some studies, thus, could reflect age-related vascular stiffening and loss of compliance (*Acierno & Labato, 2004*; *Meurs et al., 2000*) or merely an increased prevalence of diseases associated with hypertension.

Consistent with prior reports (*Acierno et al., 2018*; *Mooney et al., 2017*), we found no association between BCS and blood pressure in the mixed model analysis. The positive correlation between BCS and blood pressure results on Spearman rank analysis, thus, can be presumed to reflect confounding due to effects of subject age, weight, and anxiety score.

Blood pressure has previously been found to differ significantly among breeds, with lower blood pressures found in larger breed dogs compared to smaller breed dogs (*Höglund et al., 2012*) and published references intervals (*Bright & Dentino, 2002*). The exception to this association is sighthounds, which typically have higher blood pressures (*Acierno et al., 2018*). It has been postulated that breed associations could reflect breed-associated temperament (*Bodey & Michell, 1996*; *Chetboul et al., 2010*; *Höglund et al., 2012*; *Schellenberg, Glaus & Reusch, 2007*), but a calm external demeanor was associated with lower blood pressures among dogs of a single breed (*Bright & Dentino, 2002*). It was not possible to directly assess the impact of breed on results of this study, but weight and anxiety score were independently associated with blood pressure in this study. Taken as a

whole, these results suggest that differences in blood pressure among breeds are not entirely dependent on outward demeanor or temperament.

One concerning finding in this study was a lack of consistency in detection of hypertension. Only 7 of 30 dogs categorized as hypertensive were so categorized in both measurement periods, in spite of careful attention to optimal technique. The remainder had results consistent with hypertension in only 1 of the 2 periods. This underscores the importance of confirming abnormal results before pursuing treatment, particularly in patients without evidence of end-organ damage or diseases associated with hypertension (*Acierno et al., 2018*). Current veterinary guidelines recommend discarding the first of 5-7 indirect blood pressure readings (*Acierno et al., 2018*). However, BP 1 and BP 2-6 results did not differ in dogs when taken using either the coccygeal or radial artery and Doppler flow with headphones in one study (*Mooney et al., 2017*). Two normotensive subjects in that report had past histories of refractory hypertension due to inadequate acclimation to hospital personnel and equipment prior to blood pressure measurement. Blood pressure measurement results for both remained normal after clinical protocol was adjusted to ensure acclimation to personnel, resulting in discontinuation of antihypertensive therapy. Based on those observations, the authors recommended that acclimation of the subject to personnel and equipment be prioritized over collection of multiple readings in cases where adequate time is not available for both. Given significantly higher BP 1 vs. 2-6 results for measurements taken without headphones in this study, use of a solitary blood pressure reading only could be considered for Doppler flow blood pressure measurements taken using headphones.

The current recommendation is to use cuffs that are 30–40% of the antebrachium circumference for radial indirect blood pressure measurement (*Acierno et al., 2018*). Use of inappropriately-sized cuffs was associated with over- and underestimating indirect blood pressure when too small or too large a cuff was used, respectively, in one study of oscillometric pressure measurements in dogs (*Bodey et al., 1994*). However, a similar association was not identified in a more recent study of indirect Doppler flow collected measurements (*Mooney et al., 2017*). In this study, cuff size as a percentage of limb circumference was not correlated with blood pressure results.

This study had a few additional limitations. First, the narrow range of the anxiety scoring system used (*Scansen et al., 2014*) limited detection of smaller changes in demeanor. Furthermore, this scale hinges on factors external to the animal, including how much restraint was required and when the owner intervened—either of which could be affected by non-anxiety related factors. To better differentiate patient anxiety from non-patient factors, it is recommended that future studies evaluate anxiety using a scoring system that relies directly on changes in the patient demeanor such as the Fear and Anxiety Scoring system (*Fear Free LLC, 2017*). Additionally, the study contained few dogs with diseases or medications that could affect blood pressure results. The magnitude of the white-coat effect was significantly higher in cats with experimentally-induced kidney disease compared to healthy cats in one study (*Belew, Barlett & Brown, 1999*). Thus, caution should be exercised when extrapolating results of this study to dogs with chronic kidney disease, other diseases associated with hypertension, or receiving medications known to

alter blood pressure. Furthermore, the impact of potential habituation due to recurrent admission on results was not assessed. Both increased and decreased blood pressure were noted in healthy research cats undergoing repeated simulated office visits (*Belew, Barlett & Brown, 1999*). Similarly, both situational hypertension and masked hypertension have been identified in people (*Acierno et al., 2018*). Conversely, although there was no significant association between blood pressure results in greyhound blood donors and number of prior admissions for blood collection (*Marino et al., 2011*), oscillometric blood pressure results were significantly higher when results were taken in the hospital vs. at home, regardless of whether home measurements were collected by the investigator or the owner. A crossover design was not used for that study and details were limited regarding the blood pressure measurement protocol, including factors such as length of acclimation to the investigator in the hospital environment (the first measurement series taken). Finally, it is unclear whether the correlation analyses between blood pressure results and number of hospital visits were performed using measurements collected in hospital vs. at home. Thus, it is unclear whether the significant difference in results between locations in the study by Marino et al. was due to hospital-induced white-coat effect, inadequate initial acclimation to the investigator/protocol, or habituation to the measurement technique over time. Either of the latter possibilities could have obscured an association between habituation to the hospital and blood pressure results. Further evaluation using a crossover design might aid in clarifying the separate effects of habituation to measurement personnel vs. environment on white-coat effects in dogs.

## CONCLUSIONS

Initial indirect Doppler systolic blood pressure measurements taken without headphones were significantly higher in dogs than measurements taken using headphones. This effect did not persist for BP 2-6 measurements, which could reflect acclimation to the experience. Conversely, exclusion of 10 times as many dogs from the trial due to anxiety when measurements were first taken without headphones could have masked a significant association between BP 2-6 results and blood pressure measurement type. This possibility is supported by the strong association identified between anxiety score and blood pressure results in this study. Pending further evaluation using a larger sample size, it is prudent to use headphones for indirect systolic blood pressure measurement determination to avoid erroneous diagnosis of hypertension in dogs.

### Funding

This work was supported by the University of Tennessee College of Veterinary Medicine Center of Excellence in Livestock Diseases and Human Health Summer Research Program. There was no external funding received for this study. The funders had no role in study design, data collection and analysis, decision to publish, or preparation of the manuscript.

## Grant Disclosures

The following grant information was disclosed by the authors:

University of Tennessee College of Veterinary Medicine Center of Excellence in Livestock Diseases and Human Health Summer Research Program.

## Competing Interests

The authors declare there are no competing interests.

## Author Contributions

- India R. Gill and Jacqueline C. Whittemore conceived and designed the experiments, performed the experiments, analyzed the data, prepared figures and/or tables, authored or reviewed drafts of the paper, approved the final draft.
- Joshua M. Price analyzed the data, prepared figures and/or tables, authored or reviewed drafts of the paper, approved the final draft.

## Animal Ethics

The following information was supplied relating to ethical approvals (i.e., approving body and any reference numbers):

This study was approved by the Institutional Animal Care and Use Committee of the University of Tennessee, Knoxville (Protocol number 2428).

## Data Availability

The raw data are available in the Supplemental File.

## Supplemental Information

Supplemental information for this article can be found online at http://dx.doi.org/10.7717/peerj.7440#supplemental-information.

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
