# Peer review of "Indirect Doppler flow systolic blood pressure measurements taken with and without headphones in privately-owned, conscious dogs"

_PeerJ, doi:10.7717/peerj.7440_

## Round 0.1 · original submission · Major Revisions

Please carefully review and respond to both reviewer's concerns. Thank you very much.

Reviewer 1 ·

Basic reporting

This manuscript is well written. The language is professional throughout with minimal if any grammatical errors. The figures are relevant and easily read. The raw data is supplied and does raise some questions which will be directed in other portions of this review. The references were adequate . The introduction was adequate.

Experimental design

The research is original in nature. The questions are well defined, the main one being if determining hypertension using headphones is a more sensitive manner of blood pressure determination than without headphones. Exclusion of difficult to handle patients might have caused bias, but not clear if could be performed in any other way with out sedation of patients which would have induced additional bias.
Methods were described sufficiently.
The stats as described were a bit confusing for most general readers. These should be reviewed by a statistician and perhaps clarified in the methods section in more easily understood manner by general readers
The study does need to be expanded to increase the number of dogs and then to divide categories of co-morbidities which were not addressed at all. How might this have impacted the blood pressure, a fair number were on medications?
Your most important issue in terms of research question about headphone use is addressed, expanding numbers of cases might also help reinforce the results. However, the effect of age, weight , anxiety should be expanded upon in a more rigorous manner since you elected to include this as part of the study. In addition, you elected to include , " a typical hospital" population with a variety of presenting problems but did not address any impact that these co-morbidities or the medications used to treat these patients may have impacted the study, or induced bias. Were patients with CKD more used to being in the hospital having blood pressure determinations? If these parameters were looked at statistically and not significant would be helpful for readers to know

Validity of the findings

I thank the authors for providing raw data as well as statistical analysis.
The areas that this reviewer has the most problem is listed above and repeated here.
Your most important issue in terms of research question about headphone use is addressed, expanding numbers of cases might also help reinforce the results. However, the effect of age, weight , anxiety should be expanded upon in a more rigorous manner since you elected to include this as part of the study. In addition, you elected to include , " a typical hospital" population with a variety of presenting problems but did not address any impact that these co-morbidities or the medications used to treat these patients may have impacted the study, or induced bias. Were patients with CKD more used to being in the hospital having blood pressure determinations? If these parameters were looked at statistically and not significant would be helpful for readers to know

Additional comments

Well written.
Overall a good discussion
A study that truly needed to be performed
Most of areas of weakness or need for further study have been addresses.
No real discussion of impact of co-morbidities or medications patients receiving were discussed.
Your most important issue in terms of research question about headphone use is addressed, expanding numbers of cases might also help reinforce the results. However, the effect of age, weight , anxiety should be expanded upon in a more rigorous manner since you elected to include this as part of the study. In addition, you elected to include , " a typical hospital" population with a variety of presenting problems but did not address any impact that these co-morbidities or the medications used to treat these patients may have impacted the study, or induced bias. Were patients with CKD more used to being in the hospital having blood pressure determinations? If these parameters were looked at statistically and not significant would be helpful for readers to know

Reviewer 2 ·

Basic reporting

no comment

Experimental design

no comment

Validity of the findings

no comment

Additional comments

Review “Indirect Doppler flow systolic blood pressure measurements
taken with and without headphones in privately-owned,
conscious dogs” Peer J


General comments:
The study is well designed and carried out. I am glad such a study was done and will be a good addition to the literature on BP measuring.

Minor specific comments:
Lines 101-102: “A soft measuring tape was used to measure the circumference of the mid-antebrachium, and a cuff was selected that was 30-40% of the appendage” – this second part of the sentence is not so clearly written it should be that the width of the cuff should be 30-40 % of the limb (extremity) at the site of cuff placement. It would be more understandable to the reader.

Line 149 why authors repeat “at the start of measurements” twice, is this necessary?
Lines 178-182- the results of median measurements with or without headphones dffer from those in the table 1. Please refer to the comment bellow.
Table 1: BP should have its unit written
It is not clear: the results section BP results are written as median and range that is ok, but in the Table one it is confusing what is BP 1 – a mean +/-SD of the group using headphones and than no headphones and vice versa in the second column? And in the next row mean of BP 2-6 – the same comment.

---

## Round 0.2 · accepted · Accept

Thank you very much for your diligence in the peer-review process.

Reviewer 1 ·

Basic reporting

The authors have addressed the reveiwers comments. Well written. Complete literature review.

Experimental design

No Comment

Validity of the findings

No comment

Additional comments

Well written. Thank you for this study. The response to reviewer comments were thoughtful and well articulated

Reviewer 2 ·

Basic reporting

no comment

Experimental design

no comment

Validity of the findings

no comment

Additional comments

Reviewer 2.
I am pleased with the authors' corrections and I have no further comments.
Congratulations.